# Xylitol Fluoride Varnish: In Vitro Effect Analysis on Enamel by Atomic Force Microscopy

**DOI:** 10.3390/biomedicines10081900

**Published:** 2022-08-05

**Authors:** Catalina Iulia Saveanu, Oana Dragos, Daniela Anistoroaei, Livia Ionela Bobu, Alexandra Ecaterina Saveanu, Adina Armencia, Sorina Mihaela Solomon, Oana Tanculescu

**Affiliations:** 1Surgical Department, Faculty of Dental Medicine, University of Medicine and Pharmacy Grigore T. Popa, 700115 Iasi, Romania; 2National Institute of Research-Development for Technical Physics—IFT, 700115 Iasi, Romania; 3Faculty of Dental Medicine, University of Medicine and Pharmacy Grigore T. Popa, 700115 Iasi, Romania; 4Department of Odontology-Periodontology and Fixed Prosthodontics Faculty of Dental Medicine, University of Medicine and Pharmacy, 700115 Iasi, Romania

**Keywords:** xylitol fluoride varnish, atomic force microscopy, roughness, enamel

## Abstract

(1) Background: Numerous studies have shown the beneficial role of fluoride in the primary prevention of dental caries. The aim of the present study was to put into evidence the change in the enamel structure immediately after the application of a fluoride varnish. (2) Methods: A xylitol–fluoride varnish was evaluated. The enamel specimens (*n* = 10) were analyzed by atomic force microscopy on enamel surface and treatment with fluoride varnish applied. The dimensional topographic analysis was performed by 2D and 3D analysis software. Statistical analysis was performed using SPSS Version 26.00 (IBM, Armonk, NY, USA). A one-sample statistics test was used to identify significant differences (*p* < 0.05). (3) Results: Surface roughness (Ra) measurements ranged from Ra = 0.039 μm (±0.048), to Ra = 0.049 μm (±0.031), respectively (*p* < 0.05), with an increase in the surface roughness passing from the intact enamel to the enamel exposed to fluoride varnish. When comparing Ra values of the nonfluorinated enamel and fluorinated enamel, significant differences (*p* < 0.05) were found, suggesting that the varnish had a protective effect on the enamel surface. (4) Conclusions: Xylitol–fluoride varnish, even in one single short-time application, is effective in reducing the surface roughness of enamel structure exposed to abrasion, thus increasing resistance to dental caries.

## 1. Introduction

The oral environment has a significant influence on the hard dental tissues in the initiation and evolution of carious and non-carious lesions. Tooth enamel is a hard tissue, which contains 95% mineral substances in its composition. The mineral substances in the enamel are found in the form of calcium phosphate, HA (hydroxyapatite), FA (fluorapatite), carbonates, and silicates. When its structure is affected by various destructive processes, the enamel does not have the possibility of spontaneous regeneration [1].

For this reason, in the case of carious lesions, primary prevention is very important, and may include multiple methods. Of these, food hygiene and oral hygiene are aspects that are strictly under patient control. The application of professional fluoridation agents, however, is mainly related to the dentist.

The beneficial role of fluoride in the primary prevention of dental caries has been demonstrated by numerous studies. The main mechanisms by which fluoride fulfills this role are based on inhibiting demineralization and promoting the remineralization of dental hard tissues, as well as on the antibacterial effect of inhibiting enolase, the enzyme necessary for bacteria to metabolize carbohydrates [2,3,4,5,6]. It has been shown that, by this mechanism, fluoride varnishes can reduce the number of cariogenic bacteria (*S. mutans*) by more than ten times [7].

Fluoride can reduce coronary carious disease by up to 60% [8]. Fluoride varnishes have the advantage that they can be applied exactly in retentive areas or areas affected by demineralization. The incorporation of resins with very low viscosity in the structure of the varnish ensures its increased adhesion. Due to their content in antiseptic substances (thymol, chlorhexidine, and xylitol), the varnishes also have an effect against bacterial plaque development. In addition, another clinical advantage of applying these products is that they do not require the removal of plaque from the dental surface, which means that they can be applied very easily in a very short time. They dry very quickly and adhere to the tooth surface even in the presence of saliva. Thus, the amount of fluoride ingested is very small or even negligible, and is not a risk factor for fluorosis, a fact highlighted in specialized studies [9].

The role of fluoride varnish in carious lesions prevention was highlighted in numerous in vivo studies [10,11,12]. This method of local fluoridation, as well as gel fluoridation, is mainly recommended to children at high risk of caries [13,14,15,16]. The fluoride varnish is recommended to be applied two to four times a year, by a dentist, dental hygienist, or other health care professional, as a type of topical fluoride therapy [17]. It is also recommended in cases of dentinal hypersensitivity, in patients wearing orthodontic appliances, or to prevent cavities on exposed root surfaces [18,19].

Fluoride can cause rapid changes in the structure of the enamel when applied topically in high concentrations of 5% NaF (sodium fluoride), or 22,600 ppm fluoride [20]. Thus, calcium and phosphorus ions in the enamel, which are continuously lost during the demineralization stages, are subsequently recovered in the form of fluorapatite. Fluoride applied topically can penetrate hard dental tissues to a depth of 30 microns [21].

Enamel demineralization begins at a critical pH of 5.5. The remineralization of incipient lesions is a widely discussed process that is not yet fully known. One of the factors in-volved in remineralization is saliva, which, in addition to other important functions, such as acid neutralization, also has the function of remineralization [22]. The remineralization capacity of saliva is ensured by its content in calcium, fluoride, and phosphate ions that are naturally found in it. In the absence of ions necessary for remineralization, or as a result of repeated acid attacks (when the buffer capacity of the saliva is exceeded), the demineralization process takes place. Thus, in order to improve the remineralization capacity, the application of fluoride products becomes necessary.

One of the questions that is asked about the application of these products is what happens to the enamel and dentin and whether they can really change their structure in a positive way after a single application, given that fluoride varnishes are recommended to be applied at least once every 6 months. Profluorid varnish has a content of 5% NaF (22,600 ppm F). The fluoride ion, along with the calcium ions from the tooth structure, will precipitate calcium fluoride to form fluorapatite which is more resistant than hydroxyapatite to future acid attacks. The product also contains xylitol with an antibacterial effect. The presentation in the form of single doses makes the application very easy and eliminates the risk of overdose. The application of fluoride varnish is ideal for the prevention of dental caries, as it is resistant to moisture, is aesthetic, and comes with a high availability of flavors. Its application is also indicated in the treatment of cervical lesions and in cases of molar hypomineralization.

The application of fluoride on the enamel surface leads to minerals replacement. Specialized studies have shown that Profluorid Varnish leads to an immediate remineralization compared to other fluoride products. Once applied, the material begins to set in the presence of saliva and hardens to the tooth surface in less than 30 s. The present study aimed to investigate the result of the application of fluoride varnish product in the enamel, by analyzing the topographic structure of the enamel using atomic force microscopy (AFM) before and after the application of a varnish containing fluoride and xylitol. AFM is a technique that provides details of the analyzed surface, at the micrometer and nanometer levels. The null hypothesis is that there are no differences in the structure of the enamel, before and after fluoridation with varnish. The test hypothesis is that there are differences in the structure of the enamel, before and after fluoridation with varnish.

## 2. Materials and Methods

### 2.1. Materials

A xylitol fluoride varnish was evaluated: Profluorid Varnish (Voco) (VOCO GmbH, Cuxhaven, Germany) single dose 0.4 mL NaF 5% LOT 2109843.

### 2.2. Specimen Preparation

The enamel samples (N = 10) were prepared from freshly extracted human molars for orthodontic reasons. After extraction the teeth were disinfected in 5.25% sodium hypochlorite solution for one hour and then stored in saline solution. The specimens were sectioned longitudinally buccal–oral at the coronary level, with a high-speed tapered diamond cutter with water-air spray. Specimens with cracks, stains, or white spot lesions were excluded. The selected teeth were stored in hydrogen peroxide 1%, (pH = 5.5) at 4 °C prior to the experiment. The included specimens were embedded in a silicon cylinder with the enamel surface exposed. The specimens were ground under running water using a polishing machine with 320, 600, and 1200-grit silicone–carbide papers [23]. Thereafter, the specimens were cleaned in an ultrasonic device with deionized water for five minutes. The specimens were covered with two layers of acid-resistant nail varnish, leaving an exposed window of enamel, approximately 1 mm × 1 mm, in the center of each occlusal surface. Baseline root mean square roughness (Rrms) was measured for all the specimens before beginning the experiment, and it was observed that Rrms values of the specimens were comparable.

### 2.3. Mineralization

#### Treatment Protocols

The enamel specimens were analyzed (N = 10) based on the type of treatment used as follows: before fluoridation—intact enamel; after fluoridation—intact enamel with Profluorid varnish applied. The fluoride varnish (Profluorid Varnish) was applied in a thin layer using an applicator, according to the manufacturer’s instructions. The specimens were stored in artificial saliva at 25 °C for six hours [23,24]. The varnishes were then carefully removed with acetone solution (1:1 water) and a plastic scaler to avoid touching the enamel surface [25]. For this study it was decided to apply the varnish for 1 min to highlight the effect at a short application time.

### 2.4. Observations on AFM

The specimens were analyzed before and after the application of the fluoride varnish by AFM (Park SYSTEMS XE-100, Mannheim, Germany). The roughness was analyzed for each specimen for a size of 10 μm × 10 μm, with a scan rate of 0.5 Hz and a resolution of 256 × 256 pixels with 300 points in a row. The average surface roughness of each sample was measured with a profilometer (Mar Surf PS 1, MahrGmbH, Esslingen, Germany) for Rpv (maximum profile height and distance between the highest peak and the average roughness profile line), Rq (average deviation quadratic of the evaluated profile and represents the standard deviation of the profile height distribution), Ra (arithmetic mean deviation of the evaluated profile), and Rz (deviation on the Z axis). The dimensional topographic analysis was performed by analysis software (XEI—Image Processing and Analysis, v.1.8.0, Park Systems, Suwon, Korea) and the 3D topographic data was analyzed by using 3D non-contact optical profilometry with data analysis software (Nano Scope III, Version 5.12r2, Digital Instruments, Santa Barbara, CA, USA). The analyzes were performed at the Institute for Physical Research in Iasi, Romania.

### 2.5. Statistical Analysis

Statistical analysis was performed using SPSS for Windows, Version 26.00 (IBM, Armonk, NY, USA). A one-sample statistics test was used to identify significant differences in Rrms among the two groups. The level of significance was set at *p* < 0.05.

## 3. Results

### Analysis of the Structure of the Enamel by AFM

The analysis of the enamel structure highlights differences in its topographic characteristics. Therefore, it is possible to follow different aspects of the enamel structure depending on the scanning area and on the scanned micro zone Figure 1, Figure 2 and Figure 3. Differences can be observed between the roughness values at the enamel level as follows: on a scanning area X Scan Size 10 μm, Y Scan Size 10 μm (Figure 1); X Scan Size 5 μm, Y Scan Size 5 μm (Figure 2); and X Scan Size 1 micrometer, Y Scan Size 1 micrometer (Figure 3).

Surface roughness (Ra) measurements ranged from Ra = 0.039 μm (±0.048), Rz = 0.213 μm (±0.232), to Ra = 0.049 μm (±0.031), Rz = 0.341 μm (±0.274), respectively (*p* < 0.05), with an increase in the surface roughness passing from the intact enamel to the enamel exposed to fluoride varnish, which suggests a remineralizing effect for the varnish. Comparing Ra and Rz values of the nonfluorinated enamel and fluorinated enamel, significant differences (*p* < 0.05) were found, suggesting that the varnish had a protective effect on the enamel surface (Table 1).

The differences in values between non-fluorinated and fluorinated enamel surfaces are statistically significant (*p* < 0.05) with an increase in surface roughness from intact enamel to enamel exposed to fluoride varnish Table 2.

The roughness of the enamel structure varies in the direction of its increase in the fluoridated region at Ra = 110.8 nm and Rz = 408.53 nm from Ra = 8.13 nm and Rz = 49.287 nm before fluoridation. It can be seen that line red profile is more uniform after the application of the fluoride varnish (Figure 4B) than before its application (Figure 4A). Additionally, the fluorinated enamel surface has a less fractured appearance (Figure 4B) compared to the structural image of the initially analyzed enamel (Figure 4A). This suggests the effect on the structural change of the enamel after the application of fluorine varnish.

The same aspect can be visualized at a smaller scanning area, respectively, of 5/5 μm, where a roughness of Ra = 7.99 nm and Rz = 44.30 nm can be observed. The enamel has a fractured contour with multiple irregularities, an aspect highlighted in Figure 5A. After applying the fluorine varnish, an increase in the average roughness Ra = 49.60 nm and Rz = 158.368 nm can be observed, but an attenuation of the red profile line is seen in Figure 5B. The attenuation of the structural aspect can also be observed (Figure 5B).

Roughness analysis at a smaller scan area of 1/1 μm shows a decrease in the fluorinated surface roughness from initial values of Ra = 4.5 nm and Rz = 19.256 nm to fluorinated surface values of Ra = 3.46 nm and Rz = 14.71 nm, respectively (Figure 6A,B). This suggests a minimal modification of the line red profile without attenuating it, as found in the larger scanning areas of 5/5 and 10/10 μm. Additionally, the three-dimensional images show an attenuation of the structural morphology of the enamel after fluoridation (Figure 6B) compared to the initial structure of the enamel (Figure 6A).

## 4. Discussion

The enamel is the only hard dental tissue exposed intraorally. A decrease in the pH of the oral cavity below the critical value of 5.5 is followed by the beginning of hydroxyapatite dissolution. In the presence of fluoride, this produces the formation of fluorapatite crystals with lower solubility than that of hydroxyapatite, which reduces the process of enamel demineralization [21]. In this way, the ions of calcium and the phosphorus of the enamel that are lost during the demineralization process are recovered by remineralization as fluorapatite [26,27].

The presence of fluoride deposited as calcium fluoride on the dental surface in-creases the availability of this ion in the fluid of the biofilm. When the pH drops to a critical level, the demineralization process begins and allows the fluoride to precipitate and, thus, to restore the tooth structure [28].

In order to prevent the demineralization of the enamel and to increase its resistance against erosive attacks, topical applications of highly concentrated fluorides have been proposed in various forms of presentation: oral rinses, gels, or varnishes [24].

Compared to other fluoride vehicles, varnishes are easy to apply, safe, and well tolerated by young children and uncooperative patients. In addition, they ensure the formation of a mechanical barrier. Due to their common clinical use, the high amount of fluoride contained, and the slow release of components, varnishes can be considered the ideal products to prevent the loss of minerals from hard dental tissues [29].

In the present study, the varnish was applied only once and then completely re-moved after 6 h, to simulate the real clinical conditions, in which the varnish is subject to the risk of being removed by toothbrushing and chewing. In this way, the emphasis was more on the chemical effect and less on the mechanical protection [30].

The results of the present study showed that the local application of fluoride in the form of a NaF-containing varnish was effective in reducing the roughness of the enamel structure. Therefore, the use of this type of varnish can be a good alternative to ensure protection against changing enamel roughness and loss of hard dental structures [31].

The effectiveness of sodium fluoride in protecting the enamel against erosive attack has been demonstrated by previous studies [24,31].

This can be explained by the fact that the application of fluoride varnishes is followed by the formation of a protective layer of CaF_2_ on the surface of hard dental structures. It acts as a physical barrier, inhibiting the contact of the acid with the enamel. At the same time, this layer is also involved in the demineralization and remineralization processes: once the formation of fluoride reservoirs takes place, it participates in the inhibition of the demineralization process and in the precipitation of fluor(hydroxy)apatite [32].

Xylitol is a non-acidogenic sweetener that can form complexes with calcium ions. Due to this property, it is believed that xylitol may increase remineralization and may inhibit the dissolution of calcium and/or phosphate ions from the enamel structure. However, the role of xylitol as a preventive or remineralizing agent has not yet been sufficiently demonstrated [32,33].

The product used in the present study is a varnish containing xylitol/NaF, which produced a reduction in the roughness of the enamel and the loss of hard dental tissue after its abrasion. The values obtained in terms of surface roughness indicate that the varnishes containing this combination of xylitol and NaF are able to achieve surface protection [32].

The ability of conventional fluoride products, such as NaF varnish, to prevent erosive/abrasive processes has been described in the literature [34,35]. Regarding the presence of xylitol, it probably did not increase the protection capacity of the product against mineral losses. The results of a study conducted by Alexandria [32] indicated that fluoride- and xylitol-containing varnish reduced the enamel demineralization but not to a greater extent than xylitol-free fluoride-containing varnish.

In the present study, AFM was used as the method of analysis. AFM is a nanoindentation technique capable of obtaining images with atomic resolution after a minimum sample preparation, which is one of the main advantages of AFM compared to other analysis techniques [36]. AFM can also be used on conductive and insulating surfaces and can be performed in ambient conditions but also in air, liquids, or vacuum. Therefore, there is no risk of damage to fragile samples by harsh preparation techniques, such as coating, dehydration, or vacuum exposure, thus avoiding the artifacts associated with such techniques. In addition, AFM is a very accurate technique, which allows quantitative data to be obtained [24].

The results of the present study showed a modification in the roughness of the enamel structure after fluoridation with varnish, which means that the null hypothesis was rejected, and the test hypothesis was accepted.

These results are consistent with the results of other studies [37,38,39,40,41,42] which have shown that varnishes containing sodium fluoride are effective in modifying the roughness of the enamel surface. Other studies that have used the AFM analysis technique [42,43,44] have found that the hardness of dental hard tissues decreases after the use of fluoride varnishes. Additionally, fluoride varnish is effective in reducing enamel demineralization [45,46].

The limitations of the present study are related, in particular, to the in vitro protocol, by the inability to adequately simulate the complex biological processes that take place at the level of the teeth in the oral cavity. Another limitation is the low number of samples analyzed. Performing more tests on a larger number of samples will be able to provide more conclusive data on the analyzed aspects.

## 5. Conclusions

Xylitol–fluoride varnish, even in one single short-time application, modifies the surface roughness of exposed enamel structure. The effect of the action of fluoride applied in the form of varnish influences the superficial structure of the enamel, a fact highlighted illustratively, this being an expression of the mineralization of the structure. Even if the average surface roughness was higher after fluoridation with varnish, a fact highlighted by scanning 10/10 or 5/5 μm, where it was observed that when scanning areas smaller than 1/1 micrometer, the average surface roughness decreases. All the samples analyzed and regardless of the scanned area, the improvement of the roughness profile of the red profile was highlighted, which suggests an attenuation of the micro-roughness of the enamel.

## Figures and Tables

**Figure 1 biomedicines-10-01900-f001:**
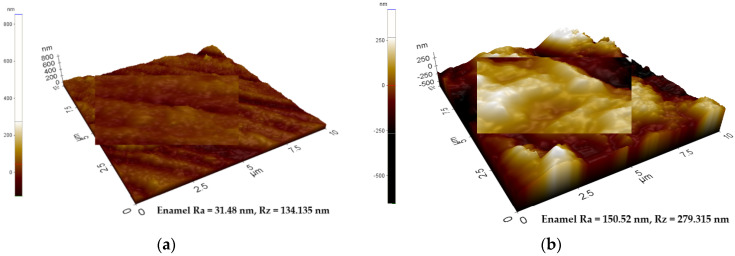
AFM image—Three-dimensional appearance of the enamel structure. The variability of the roughness profile can be observed at the enamel level. (**a**) X Scan Size 10 μm, Y Scan Size 10 μm, Ra = 31.48 nm, Rz = 134.135 nm; (**b**) X Scan Size 10 μm, Y Scan Size 10 μm, Ra = 150.52 nm, Rz = 279.315 nm.

**Figure 2 biomedicines-10-01900-f002:**
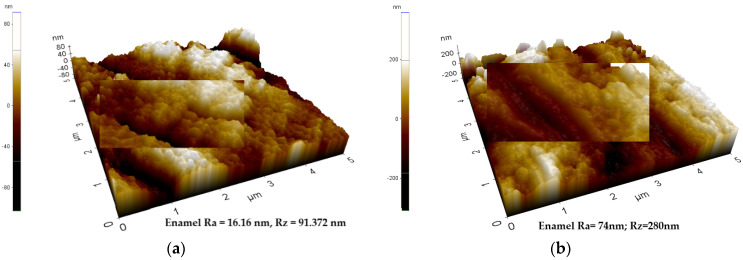
AFM image—Three-dimensional appearance of the enamel structure. The variability of the roughness profile can be observed at the level of enamel. (**a**) X Scan Size 5 μm, Y Scan Size 5 μm, Ra = 16.16 nm, Rz = 91.372 nm; (**b**) X Scan Size 5 μm, Y Scan Size 5 μm, Ra = 74 nm, Rz = 280 nm.

**Figure 3 biomedicines-10-01900-f003:**
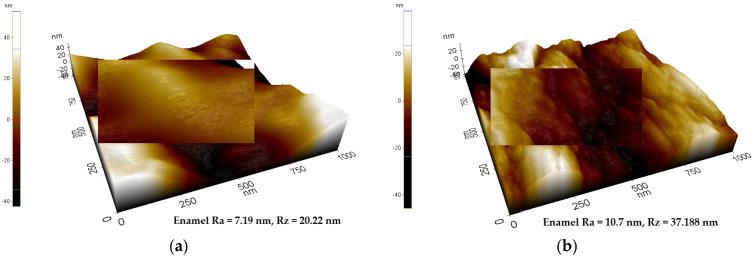
AFM image—Three-dimensional appearance of the enamel structure. The variability of the roughness profile can be observed at the enamel level. (**a**) X Scan Size 1 micrometer, Y Scan Size 1 micrometer, Ra = 7.19 nm, Rz = 20.22 nm; (**b**) X Scan Size 5 μm, Y Scan Size 5 μm, Ra = 10.7 nm, Rz = 37.188 nm.

**Figure 4 biomedicines-10-01900-f004:**
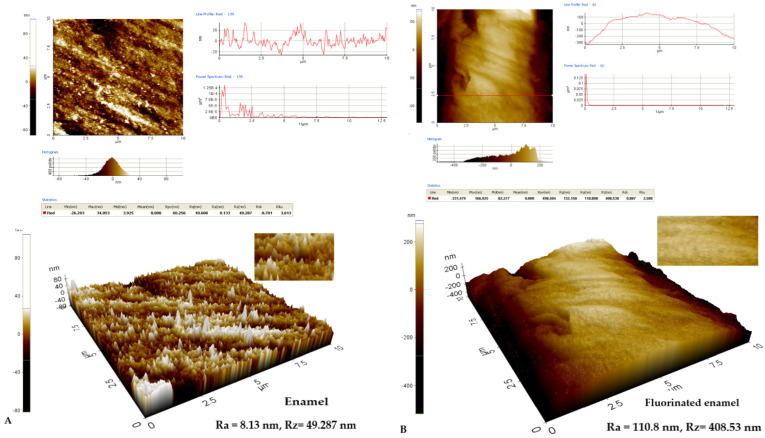
AFM images from the enamel surface layer on the X Scan layer 10 μm/Y Scan layer 10 μm: (**A**) Two-dimensional and three-dimensional AFM image of the enamel structure in the enamel without fluoridation Ra = 8.13 nm and Rz = 49.287 nm; (**B**) Two-dimensional and three-dimensional AFM image of the enamel structure in the enamel after fluoridation with varnish Ra = 110.8 nm and Rz = 408.53 nm.

**Figure 5 biomedicines-10-01900-f005:**
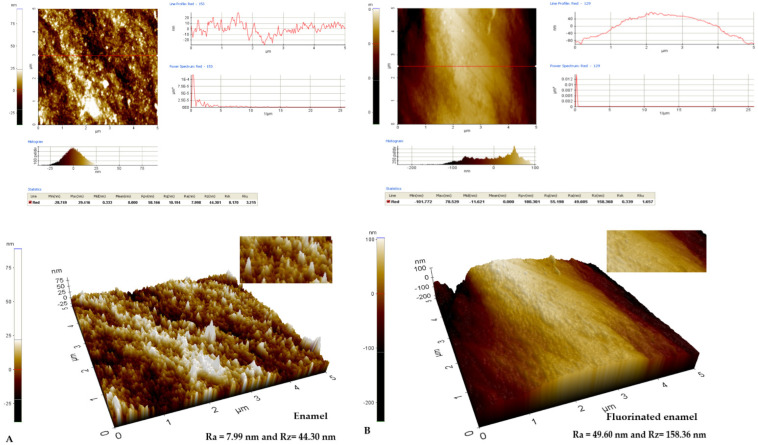
AFM images from the enamel surface layer on the X Scan layer 5 μm/Y Scan layer 5 μm: (**A**) Two-dimensional and three-dimensional AFM image of the enamel structure in the enamel without fluoridation Ra = 7.99 nm and Rz = 44.30 nm; (**B**) Two-dimensional and three-dimensional AFM image of the enamel structure in the enamel after fluoridation with varnish Ra = 49.608 nm and Rz = 158.368 nm.

**Figure 6 biomedicines-10-01900-f006:**
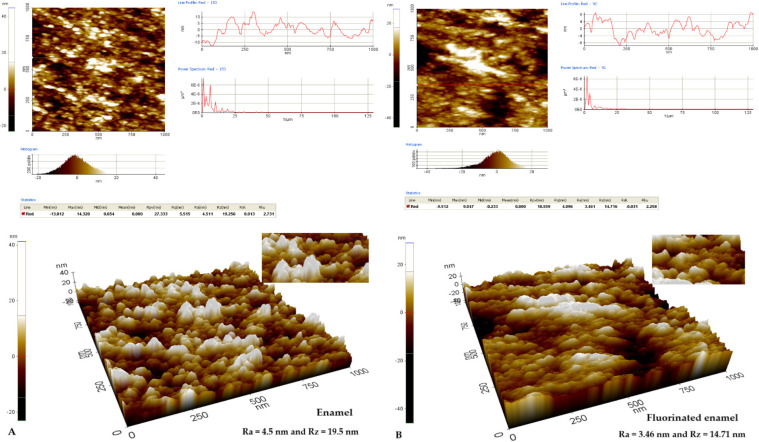
AFM images from the enamel surface layer on the X Scan layer 1 micrometer/Y Scan layer 1 micrometer: (**A**) Two-dimensional and three-dimensional AFM image of the enamel structure in the enamel without fluoridation Ra = 4.5 nm and Rz = 19.256 nm; (**B**) Two-dimensional and three-dimensional AFM image of the enamel structure in the enamel after fluoridation with varnish Ra = 3.46 nm and Rz = 14.71 nm.

**Table 1 biomedicines-10-01900-t001:** Descriptive Statistics for samples scanned on an area of X Scan size 10 μm on Y Scan size 10 μm before fluoridation and after fluoridation. The values are presented in micrometers.

Descriptive Statistics					
	N	Minimum	Maximum	Mean	Std. Deviation
Ra before	10	0.008	0.150	0.039	0.048
Ra after	10	0.007	0.111	0.049	0.031
Rz before	10	0.037	0.830	0.213	0.232
Ra after	10	0.003	0.950	0.341	0.274
Valid N (listwise)	10				

**Table 2 biomedicines-10-01900-t002:** One-sample test for samples scanned on an area of X Scan size 10 μm and Y Scan size 10 μm before fluoridation and after fluoridation. The values are presented in micrometers.

One-Sample Test						
	Test Value = 0					
	t	df	Sig. (2-Tailed)	Mean Difference	95% Confidence Interval of the Difference	
					Lower	Upper
**Ra before**	2.592	9.000	0.029 *	0.039	0.005	0.074
**Ra after**	4.995	9.000	0.001 *	0.049	0.027	0.072
**Rz before**	2.904	9.000	0.017 *	0.213	0.047	0.379
**Ra after**	3.946	9.000	0.003 *	0.341	0.146	0.537

* refers to Significance level.

## Data Availability

The data that support the findings of this study are available on request from the corresponding author.

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
