# Peer review of "Xylitol Fluoride Varnish: In Vitro Effect Analysis on Enamel by Atomic Force Microscopy"

_biomedicines, 2022, doi:10.3390/biomedicines10081900_

Round 1

Reviewer 1 Report

This is a sound experimental study on the effect of xylitol fluoride varnish on enamel in vitro. Interestingly, a single exposure to the varnish causes significant effects on the roughness of the enamel.  Unfortunately, I am not clear what the effect is, and authors appear to contradict themselves.  In line 311, they clearly state that the was a "reduction in roughness" after treatment with varnish.  Then in line 326, they state that the varnish "increases the surface roughness..."  Which is it?  I found this very confusing.

The authors really need to make the results clearer.

There are also some other corrections needed, as follows:

Change the sentence on lines 33 and 34 to: "The enamel specimens (n = 10) were analyzed by atomic force microscopy on enamel surface and treatment with fluoride varnish applied."

Line 101 is confusing, because calcium fluoride is a substance in its own right.  So I would prefer this to read: "... precipitate calcium and fluoride ions along with phosphate ions to form fluorapatite..."

Lines 115 and 116: Change "... metric..." to "... metre...". i.e. micro-metre.

Line 122: Add a spce to make it NaF 5%.

Line 130: "silicone" not "silicon"

Line 256: Spelling of the element should be "phosphorus".

Lines 309, 310: Change the final sentence to: "In addition, AFM is a very accurate technique, which allows quantitative data to be obtained."

Line 326: "increase" should be "increases".

Please check the references: In most of them, the journal titles do not have capital letters, which is incorrect.

Author Response

Response to Review

Dear reviewer,

First of all, thank you for the evaluation. I made the changes as you suggested.

Point 1

  • In line 311, they clearly state that the was a "reduction in roughness" after treatment with varnish.  Then in line 326, they state that the varnish "increases the surface roughness..."  Which is it?  I found this very confusing. The authors really need to make the results clearer.

Reponse 1:

  • I explain this in conclusion” Even if the average surface roughness was higher after fluoridation with varnish, a fact highlighted by scanning 10/10 or 5/5 micrometers, it was observed that when scanning areas smaller than 1/1 micrometer the average surface roughness decreases.”
  • I change for line 311 The results of the present study showed a modification in the roughness of the enamel structure after fluoridation with varnish, which means that the null hypothesis was rejected, and the test hypothesis was accepted.
  • I change line 326” Xylitol-fluoride varnish, even in one single short time application, modifies the surface roughness of exposed enamel structure.”

Point 2

  • Change the sentence on lines 33 and 34 to: "The enamel specimens (n = 10) were analyzed by atomic force microscopy on enamel surface and treatment with fluoride varnish applied."

Reponse 2 :

  • I changed the phrase. Thanks for the suggestion.

Point 3

  • Line 101 is confusing, because calcium fluoride is a substance in its own right.  So I would prefer this to read: "... precipitate calcium and fluoride ions along with phosphate ions to form fluorapatite..."

Reponse 3 :

  • Line 101 - I change. Thank you for your remark.
  • ”The fluoride ion, along with the calcium ions from the tooth structure, will precipitate calcium and fluoride ions along with phosphate ions to form fluorapatite which is more resistant than hydroxyapatite to future acid attacks.”

Point 4

  • Lines 115 and 116: Change "... metric..." to "... metre...". i.e. micro-metre.

Reponse 4 :

  • Line 115, 116 - I change

AFM is a technique that provides details of the analyzed surface, at the micro-meters and nanometers level.

Point 5

  • Line 122: Add a spce to make it NaF 5%.

Reponse 5 :

  • Line 122 – I made change

Point 6

  • Line 130: "silicone" not "silicon"

Reponse 6 :

  • Line 130 – I made change

Point 7

  • Line 256: Spelling of the element should be "phosphorus".

Reponse 7 :

  • Line 256 – I made change

Point 8

  • Lines 309, 310: Change the final sentence to: "In addition, AFM is a very accurate technique, which allows quantitative data to be obtained."

Reponse 8 :

  • Line 309, 310 – I made change

Point 9

  • Line 326: "increase" should be "increases".

Reponse 9 :

  • Line 326 – I made change ”Xylitol-fluoride varnish, even in one single short time application, modifies the surface roughness of exposed enamel structure”

Point 10

  • Please check the references: In most of them, the journal titles do not have capital letters, which is incorrect.

Reponse 10 :

  • Thanks, I made the changes.

Reviewer 2 Report

The objective of  this study is to investigate the immediate change in the enamel structure after the application of a fluoride varnish. It's relevant and interesting. The topic is original but is not novel. Compared with other published material, this study provide additional information on enamel morphology after fluoride application.

Overall, the article is well-written, clear and readable, and the findings are well-documented to support the concluding section.

Author Response

Response to Review

Dear reviewer,

Thank you for the evaluation. I made some changes to improve the article as follows.

1 -   I change for line 311 The results of the present study showed a modification in the roughness of the enamel structure after fluoridation with varnish, which means that the null hypothesis was rejected, and the test hypothesis was accepted.

2 I change line 326” Xylitol-fluoride varnish, even in one single short time application, modifies the surface roughness of exposed enamel structure.”

  • I changed the phrase lines 33 and 34 to: "The enamel specimens (n = 10) were analyzed by atomic force microscopy on enamel surface and treatment with fluoride varnish applied."
  • Line 101 - I change” The fluoride ion, along with the calcium ions from the tooth structure, will precipitate calcium and fluoride ions along with phosphate ions to form fluorapatite which is more resistant than hydroxyapatite to future acid attacks.”
  • Line 115, 116 - I change AFM is a technique that provides details of the analyzed surface, at the micro-meters and nanometers
  • Line 122 – I made a space NaF 5%.
  • Line 130 – I made change "silicone" not "silicon"
  • Line 256 – I made change "phosphorus".
  • Line 309, 310 – I made change "In addition, AFM is a very accurate technique, which allows quantitative data to be obtained."
  • I made the changes to the references. I put the capital letters for journals

Round 2

Reviewer 1 Report

The authors have made all the necessary changes, and the paper is now suitable for publication.  I consider this to be a very worthwhile paper, and one that makes a distinct contribution to knowledge.